# The CAR–mRNA Interaction Surface Is a Zipper Extension of the Ribosome A Site

**DOI:** 10.3390/ijms23031417

**Published:** 2022-01-26

**Authors:** Carol Dalgarno, Kristen Scopino, Mitsu Raval, Clara Nachmanoff, Eric D. Sakkas, Daniel Krizanc, Kelly M. Thayer, Michael P. Weir

**Affiliations:** 1Department of Biology, Wesleyan University, Middletown, CT 06459, USA; cdalgarno@wesleyan.edu (C.D.); kscopino@wesleyan.edu (K.S.); mraval@wesleyan.edu (M.R.); cnachmanoff@wesleyan.edu (C.N.); esakkas@wesleyan.edu (E.D.S.); 2Department of Mathematics and Computer Science, Wesleyan University, Middletown, CT 06459, USA; dkrizanc@wesleyan.edu (D.K.); kthayer@wesleyan.edu (K.M.T.); 3College of Integrative Sciences, Wesleyan University, Middletown, CT 06459, USA; 4Department of Chemistry, Wesleyan University, Middletown, CT 06459, USA

**Keywords:** ribosome translocation, molecular dynamics, CAR interaction surface, A-site decoding center, ribosome substates

## Abstract

The ribosome CAR interaction surface behaves as an extension of the decoding center A site and has H-bond interactions with the +1 codon, which is next in line to enter the A site. Through molecular dynamic simulations, we investigated the codon sequence specificity of this CAR–mRNA interaction and discovered a strong preference for GCN codons, suggesting that there may be a sequence-dependent layer of translational regulation dependent on the CAR interaction surface. Dissection of the CAR–mRNA interaction through nucleotide substitution experiments showed that the first nucleotide of the +1 codon dominates over the second nucleotide position, consistent with an energetically favorable zipper-like activity that emanates from the A site through the CAR–mRNA interface. Moreover, the CAR/+1 codon interaction is affected by the identity of nucleotide 3 of +1 GCN codons, which influences the stacking of G and C. Clustering analysis suggests that the A-site decoding center adopts different neighborhood substates that depend on the identity of the +1 codon.

## 1. Introduction

Protein translation is controlled by multiple mechanisms, providing overlapping layers of regulation that operate at the levels of translation initiation, elongation, and termination. Phosphorylation of eIF2 and eEF2 regulates initiation and elongation, respectively [1,2,3]. In addition, regulated modifications of tRNA, rRNA, and mRNA nucleotides can affect these translation stages and influence translation fidelity and efficiency [4,5,6,7,8,9]. Interestingly, K63 polyubiquitination of solvent exposed ribosome proteins can halt translation [10,11] and translation is also modulated through ribosome biogenesis [12]. In recent years, increasing evidence has emerged suggesting that alternative versions of ribosomes with different translation properties are made under different cellular conditions, such as stress [13,14,15,16]. Alternative versions of ribosomes arise through several mechanisms: They may lack or have different isoforms of ribosomal proteins or they can have differentially regulated residue modifications in their ribosomal proteins or rRNAs. Integrating these observations with our structural understanding of ribosome function presents an exciting challenge.

Using molecular dynamic (MD) methods, we recently identified a ribosome interaction surface termed CAR (Figure 1), which we hypothesize provides a layer of translation regulation under stress conditions through the production of ribosomes with an altered version of the CAR surface [17,18,19]. Protein translation requires efficient translocation of mRNA codons through the decoding center aminoacyl (A) site, where the codon base pairs with the anticodon of the tRNA that brings in the next amino acid for the growing newly synthesized protein. We hypothesize that the interaction of CAR with the mRNA helps in modulating this translocation. The CAR interaction surface consists of three residues: Nucleotides C1274 and A1427 of the *S. cerevisiae* 18S rRNA, which we will refer to as C1054 and A1196, the corresponding nucleotides in *E. coli* 16S rRNA, and R146 of ribosomal protein Rps3 [19]. C1054 and A1196 are well conserved in 18S and 16S rRNAs. However, R146 is only conserved in eukaryote ribosomes, suggesting that prokaryotes may have a partial version of the CA(R) surface [19]. Integrity of the CAR interface relies on stacking interactions, namely base stacking between C1054 and A1196, and pi–cation stacking between the A1196 base and the guanidinium group of R146. The CAR surface resembles an extension of the A-site tRNA anticodon (Figure 1) and is anchored to the anticodon through a base stacking interaction between C1054 and tRNA nucleotide 34, the anticodon nucleotide that base pairs to the wobble nucleotide of the codon.

Immediately adjacent to the A-site tRNA anticodon, the CAR surface is positioned facing the +1 codon of the mRNA—the codon that is next in line to enter the ribosome A site—allowing for the CAR surface to have sequence-dependent H-bond interactions with the +1 codon nucleotides (Figure 1). When the +1 codon is GCU, the first nucleotide (+1 G1) base pairs with C1054 through Watson–Crick edges of both bases, and the second nucleotide (+1 C2) H-bonds with A1196 and R146. In addition, +1 C2 interacts with A1196 through a Watson–Crick/Hoogsteen interaction and R146 through H-bonding with the planar guanidinium group of the arginine. The interaction between CAR and the +1 codon evolves dynamically during the various stages of ribosome elongation [20] and is most pronounced at translocation stage II [19], the stage analyzed in the current study.

R146 is post-translationally modified through methylation of its guanidinium group, which causes the H-bonding interaction between CAR and the +1 codon to be significantly reduced [18]. We hypothesize that ribosomes made under stress conditions have a CAR interface with an unmethylated R146, thereby potentiating the interaction between CAR and the +1 codon and permitting CAR-mediated regulation of translation [18]. Consistent with this model, the expression levels of Sfm1, the methyltransferase enzyme responsible for methylating R146 [21,22], are downregulated under stress conditions [18,23], although this may be due in part to the depression of translation levels observed under stress [24].

We propose that energetically favorable interactions between CAR and the +1 codon are highly sequence specific, implying that CAR regulation of translation depends on the codon content of open reading frames (ORFs). Indeed, highly expressed genes have particularly high levels of GCN codons in the initial codons of their ORFs (ramp regions [25]) and it has been suggested that codons in the ramp region might be particularly influential in controlling translation rates [17,25], perhaps analogous to cars accelerating onto the on-ramp of a highway. In our initial studies, we investigated the effects of changing the second nucleotide of +1 codons and showed that the CAR/+1 codon interaction is strongest with GCU codons [19]. In this study, we have expanded this analysis and found that +1 GCU codons have the highest levels of H-bonding with CAR, and that the nucleotide at position 1 of the +1 codon is more influential than position 2 in determining the strength of the CAR–mRNA interaction, similar to a zipper extension of the A-site codon/anticodon base pairing. In addition, the nucleotide identity of position 3 modulates stacking of the other two +1 codon nucleotides and their interaction with CAR. Our structural analysis of the A-site neighborhood suggests that, in each of its codon steps during ribosome translocation, the decoding center neighborhood adopts different substates depending on the identity of the +1 codon. The sequence-dependent modulation of CAR–mRNA interactions may impact translation speeds and protein production levels in response to stress conditions.

## 2. Results and Discussion

Sequence specific interactions of CAR with the +1 codon, the codon next to enter the ribosome A site, are likely to be important for regulation of protein translation. To investigate energetically favorable interactions, we developed an information theoretic framework to compare different +1 codons and assess the relative importance of different nucleotide positions in the codon.

### 2.1. A Ramp Codon Position Weight Matrix Reveals Nucleotide Selection Preferences

Individual information scores [26] of codons were used to guide our assessment of the behaviors of different codons. Individual information scores, which can reflect the free energies of a sequence motif’s ligand interactions [27], are calculated based on the deviations of nucleotide frequencies from their expected background frequencies. Sequence patterns that have higher than expected frequencies may have selective advantages that often reflect biological functions. In addition, patterns with lower than expected frequencies may have selective disadvantages. We hypothesized that these selective pressures on codons might result in part from interactions of +1 codons with the CAR interface, although of course many other codon properties, such as tRNA availability [28] might also contribute to differences in selective pressure.

We computed a position weight matrix to score the individual information of codons in the ramp regions of ORFs (Figure 2A). We focused on codons 3–7 of ORFs (6033 verified and uncharacterized *S. cerevisiae* genes; https://www.yeastgenome.org, accessed on on 2 May 2021) since these codons exhibit particularly strong GCN periodicity in highly expressed genes [17,29,30]. In addition, we suspect this ramp region may be particularly influential in the hypothesized CAR-mediated regulation of translation [17,19]. By superimposing codons 3–7, we created a meta-codon representation of these codons examining nucleotide choices at the 1st, 2nd, and 3rd positions of each meta-codon. Each of the three nucleotide positions were treated as independent and the codon position weights were calculated using the background frequencies in ORFs (A: 0.328, C: 0.192, G: 0.204, T: 0.276). Addition of the position weights, log_2_(f_observed_/f_expected_) across positions 1, 2, and 3 of each codon sequence provided individual information scores (Figure 2B).

Since individual information scores reflect deviations from expected frequencies with potential functional correlates, we were interested in how trends in individual information scores might relate to the CAR function measured by monitoring H-bonding of CAR to +1 codon as well as the structural integrity of CAR. Comparisons of codon scores (Figure 2B) showed that GCU has the highest individual information, indicating the highest positive deviation from the expected frequencies. We hypothesize that this may be related in part to the CAR function, given that the +1 GCU structure aligns with and exhibits significant H-bonding to CAR (Figure 1). In addition, we were curious to compare the CAR function for +1 codons over a range of individual information scores. Our previous testing [19], starting with +1 GCU and varying position 2, showed that +1 GCU has the highest H-bonding and +1 GGU has the lowest, similar to the individual information trend. In the present study, we wished to expand this analysis to compare A and G at position 1 (+1 GNN cf. +1 ANN) since these codons have similar individual information, but might have different CAR behaviors. Similarly, we wished to find out whether +1 codons starting with U or C (+1 UNN and +1 CNN) have characteristic CAR behaviors, given that their individual information scores tend to be lower and potentially detrimental. We decided to limit our initial analysis to +1 codons with position 3 fixed at U (NNU, Figure 2C), given that our previous testing [19] suggested that positions 1 and 2 are likely to be the dominant interactors with CAR (Figure 1).

### 2.2. CAR–mRNA H-Bonding Is Sequence Dependent

MD simulations with the AMBER suite [31] were used to assess CAR interactions with the +1 codon, as described previously [18,19]. We used a 494-residue subsystem of the ribosome centered around C1054 of the CAR interface that included the A-site codon and +1 codon of the mRNA, and an anticodon region of the tRNA. The residues at the edges of the subsystem were restrained to their initial coordinates (onion shell [18,19]) to conform to translocation stage II (PDB ID 5JUP [20]), the stage that exhibits the most robust CAR/+1 codon H-bond interactions.

As discussed above, our previous analysis [19] showed that the +1 GCU codon interacts with the CAR interface dominated by H-bond interactions between +1 G1 and C1054 of CAR, and +1 C2 with A1196 and R146 of CAR. Minor interactions were also observed between +1 G1 and A1196 as well as +1 G1 with nt34 of the tRNA, which normally H-bonds with the A-site wobble nucleotide. In addition, +1 C2 had minor interactions with C1054. These minor interactions are included in the average H-bond counts in Figure 3A.

Our overall results showed that +1 GCU has the highest H-bonding to CAR compared to the other tested +1 codons (Figure 1, Figure 3 and Appendix A). CAR H-bond levels are lower when +1 G1 or C2 is replaced, and these reductions are associated with elevated H-bonding of the A-site wobble nucleotide with tRNA nt34. Replacement of +1 G1 or C2 also affects anchoring of CAR to the tRNA and stacking interactions of the CAR interface, as discussed later.

The levels of H-bond interactions of the +1 codon with CAR can be grouped into a tiered series (Figure 3B): *High* (GCU), *intermediate* (G[GAU]U, CCU, UNU), and *low* (ANU, C[GAU]U). The *high* group has optimal nucleotides at both positions 1 and 2 of the +1 codon (+1 G1 and +1 C2); the *intermediate* group has optimal nucleotides at one of the two positions (+1 G1 or +1 C2) or UGN as discussed below; the *low* group has suboptimal nucleotides at positions 1 and 2 (not +1 G1 and not +1 C2).

The UNU codons are qualitatively different from the other intermediate group codons as the +1 U1 interaction is split evenly between C1054 and A1196 (Figure 4 and Appendix A). The prominent off-phase U1/A1196 may be associated with properties that are selected against (see below). In contrast, +1 G1 shows a very strong H-bonding preference for C1054, whereas +1 C1 and +1 A1 H-bond at similarly low levels to the three residues: tRNA nt34, C1054, and A1196 (Figure 4).

Overall, these results suggest that the ribosome’s CAR surface has sequence specific interactions with codons of the mRNA ORF when they are in the +1 position.

### 2.3. CAR Behaves as an Extension of the A-Site Anticodon

Nucleotide substitutions of the optimal +1 GCU result in reduced H-bonding between CAR and the +1 codon. These reductions show a graded effect from 5′ to 3′ in the +1 codon, suggesting a zipper-like behavior. If the +1 G1 interaction with CAR is disrupted by replacing G1, this leads to significant reductions in H-bonding of both positions 1 and 2 of the +1 codon (Figure 5A). However, if the +1 C2 interaction is disrupted, this results in significant H-bond reductions of only position 2, whereas position 1 has minor reductions (Figure 5B). This suggests that interactions between the +1 codon nucleotides and CAR are facilitated by interactions on their 5′ side, suggesting a zipper-like effect emanating from the A site through the CAR/+1 codon interaction (Figure 1).

Although nucleotide 3 of the +1 codon does not interact directly with the CAR interface, replacing U3 in +1 GCU reduces C2 H-bonding with CAR (Figure 5C), suggesting that the third nucleotide of the +1 codon can influence the positioning of C2. Consistent with this, examination of G1:C2 base stacking (Figure 5D and Appendix A) showed that the alignment of the stacked base rings was less centered and more variable when U3 was replaced with other nucleotides. Interestingly, codons with U at position 3 have higher individual information scores than other third position nucleotides (Figure 2B).

The above analysis suggests the dominance of position 1 over position 2 in determining the extent of H-bonding of CAR to the +1 codon. Therefore, to compare the CAR interactions of codons with different individual information scores, we grouped our data according to which nucleotide was at position 1. Consistent with the apparent zipper behavior of the CAR/+1 codon interaction, we observe a correlation of codon individual information with the strength of the CAR/+1 codon interaction, but only with optimal nucleotide choices at position 1 of the +1 codon (G1 and to some extent C1; Figure 6B). For this analysis, we compared individual information scores of codons with the sum of the H-bond interactions of nucleotides 1 and 2 with CAR (and nt 34; Figure 6A). We found that with G1 (+1 GNU) or C1 (+1 CNU), there is a correlation between the level of H-bonding and individual information scores of codons (correlation coefficient = 0.818; Figure 6B). Elevated individual information scores reflect selection away from the expected background frequencies, and improved CAR/+1 codon H-bonding may contribute in part to this selection suggesting possible benefits of CAR function.

However, for codons with A1 (+1 ANU) which have low CAR H-bonding (Figure 3) or U1 (+1 UNU) which have incorrectly phased H-bonding (Figure 4), we found a negative correlation of individual information scores of codons with the strength of their CAR H-bonding (correlation coefficient = −0.833; Figure 6C). Interestingly, ANU codons have individual information scores only a little below GNU (Figure 2B,C), suggesting that there may also be selection for codons (+1 ANU) that have consistent weak interactions with CAR. In contrast, UNU codons have some of the lowest individual information scores (Figure 2B,C), suggesting that there may be selection against the split or out-of-phase interactions of U1 with CAR (Figure 4). We speculate that this behavior could influence probabilities of frameshift events during translocation, and interestingly, U at position 1 is consistent with the 7-nucleotide slippery sites associated with frameshifting events if the slippery site encompassed the A site and +1 codons [33].

### 2.4. The mRNA Interactions Modulate CAR Integrity and Anchoring

Stacking interactions can make important contributions to the stabilities of RNA structures [34]. In MD simulations of the +1 GCU ribosome subsystem, as discussed above, the CAR interaction surface is anchored to the tRNA nt34 through base stacking with C1054, and the CAR surface itself displays stacking interactions between its adjacent residues: Base stacking between C1054/A1196 and pi−cation stacking between A1196/R146 [18,19]. We were curious whether +1 codons with weaker H-bonding to CAR influenced the CAR anchoring or stacking interactions.

Stacking between pairs of residues was analyzed by measuring the distance between the center of mass (COM) of adjacent residues, as described previously [18,19]. In general, COM distances between tRNA nt35 and nt34, and between C1054 and A1196, were not affected by the presence of different +1 codons (Figure 7A). However, the stacking interaction between tRNA nt34 and C1054 was stronger (smaller COM distances; Figure 7A) for +1 codons that exhibit lower CAR/+1 codon H-bonding (Figure 3). Indeed, +1 CUU and +1 UUU showed significantly lower COM distances compared to +1 GCU, indicating improved base stacking. In contrast, the stacking interaction between A1196 and R146 was less strong (higher COM distances; Figure 7A) for +1 codons that show weaker CAR/+1 codon H-bonding compared to +1 GCU. Consistent with its reduced pi−cation stacking, R146 exhibited more variability in its positioning as revealed by its elevated root mean squared fluctuations (RMSF; Figure 7B). The RMSF measurements were conducted using the four core heavy atoms of the arginine guanidinium group (ref. [18] and Appendix A). RMSF assessment of neighboring residues showed no significant differences when comparing the +1 GCU and +1 CGU subsystems (Appendix A).

We also measured the solvent accessible surface area (SASA [18,35,36]) between stacked residues comparing +1 GCU and +1 CGU as a test case (Figure 7C). The SASA measurement, comparing isolated stacked residue pairs, allowed us to assess the ability of water molecules to fit between the stacked residues. Compared to +1 GCU, +1 CGU showed greater SASA between R146 and A1196, confirming that their pi−cation stacking is less robust. In contrast, the +1 CGU subsystem showed reduced SASA between C1054 and tRNA nt34, indicating stronger anchoring of CAR.

In summary, these results showed that stronger H-bond interactions between CAR and the +1 codon are associated with stronger stacking between A1196 and R146, suggesting that the CAR/+1 codon interaction tends to stabilize the stacking interaction. In contrast, the CAR/+1 codon interaction tends to destabilize the anchoring of CAR to tRNA nt34, as evidenced by the weakened stacking between C1054 and tRNA nt34, suggesting that anchoring dominates when CAR is not held in place well by the interaction with the +1 codon.

### 2.5. Different +1 Codons Exhibit Different Substates of the Decoding Center

Our observations that different +1 codons have effects on H-bonding between CAR and the +1 codon, as well as the anchoring and stacking integrity of the CAR interface suggested that different +1 codon identities may lead to alternative dynamic substates of the ribosome decoding center. To investigate this further, we performed K-means clustering analysis [37] to determine whether different +1 codons are associated with different clusters representing alternative dynamic structures.

For the K-means analysis, we compared +1 GCU and +1 CGU subsystems. Sixty concatenated trajectories (30 for each +1 codon, 6400 frames total) were subjected to K-means clustering using the summation of RMSD measurements of backbone atoms as the difference measure for clustering. Unexpectedly, setting K = 5 resulted in a complete separation of the +1 GCU and +1 CGU trajectories (Figure 8A), indicating that these two +1 codons are associated with distinct dynamic backbone substates that extend beyond the +1 codon. Increasing the number of clusters (K = 8) revealed additional minority clusters, but left over 89% of the frames in the two dominant clusters. Decreasing the number of clusters (K <5) led to a large cluster with over 95% of the frames from both dominant clusters of K = 5.

We carried out a second K-means analysis, in which all atoms of only 12 residues—the A site and +1 codon, CAR and the tRNA anticodon—were used in the clustering steps. Once again, the two dominant clusters (K = 2) showed almost complete separation of the GCU and CGU frames. The centroid structures for the two clusters represent the structures that are closest to all of the other structures in their respective clusters (Figure 8B). The centroid for GCU has poor anchoring of the CAR interface consistent with our earlier results. In contrast, the +1 CGU centroid shows good stacking alignment of C1054 with tRNA nt34 providing the anchoring of CAR.

A comparison of the centroid structures for clusters 1 (+1 GCU) and 2 (+1 CGU) in Figure 8A showed that the differences between their two structures extend well beyond the CAR residues (Figure 8C). This was demonstrated by performing RMSF analysis on the cluster 2 trajectories using the cluster 1 centroid as a reference structure. High RMSF values (reds and yellows in heat map; Figure 8C) were detected throughout the subsystem of 321 unrestrained residues (inside the onion shell).

As summarized in Figure 3B, +1 GCU has the highest levels of H-bonding between CAR and the +1 codon, while +1 CGU is in a group of codons with the lowest H-bonding levels. We expanded our K-means analysis to include +1 codons with intermediate levels of H-bonding (+1 CCU, +1 GGU, and +1 UGU) as well as +1 AAU, which similar to +1 CGU has low H-bonding (Figure 9). Thirty trajectories for each of the six +1 codons were concatenated for K-means analysis. Clustering using backbone atoms of the 321 unrestrained residues again revealed a striking separation of high and intermediate interactors (+1 GCU, +1 CCU, +1 GGU, and +1 UGU) from the low interactors (+1 CGU and +1 AAU). This indicates a strong preference of the ribosome subsystems for these two dominant, yet distinctive backbone substates of the decoding center neighborhood. One substate is associated with CAR interactions with the +1 codon, and the other substate is associated with low interactions. Depending upon which codon has advanced to the +1 position during translocation (stage II), the decoding center would switch between these two substates. These results emphasize again the striking neighborhood effects of the +1 codon.

## 3. Conclusions

In this study, we have shown that the identity of the +1 codon, the codon next to enter the ribosome A site, has a pronounced effect on the behavior of the decoding center structure and dynamics. This effect is driven by whether or not the +1 codon interacts with the CAR interface, a surface that shows sequence-specific interactions with the +1 codon. CAR exhibits a strong preference for +1 GCN codons, resulting in switch-like behavior of the ribosome subsystem depending on whether or not the codon conforms to GCN. The consequences of GCN codon interactions with CAR on protein translation remain to be elucidated, although preliminary indirect observations suggest that energetically favorable CAR/+1 codon interactions with initial codons of protein ORFs (ramp regions) may be particularly influential in modulating protein translation under certain cellular conditions, such as stress when CAR is hypothesized as functional [17,18,19]. It is interesting to speculate that over evolutionary timescales, the selection for GCN codons in the ramp regions of ORFs may have rendered genes sensitive to CAR-mediated regulation. GCN codes for alanine, and the small non-polar R group of alanine may be expected to often have fairly neutral effects on the structure and function of evolving proteins, if the selection for GCN led to the incorporation of alanine residues into the protein structure.

## 4. Materials and Methods

### 4.1. Information Theoretic Analysis of Codons

Python scripts were used to analyze yeast ORF sequences of 6033 verified and uncharacterized genes (downloaded on 2 May 2021; https://www.yeastgenome.org). For calculations of individual information scores [26], background nucleotide frequencies were calculated for all ORFs (A: 0.328, C: 0.192, G: 0.204, T: 0.276). Individual information scores were calculated using a position weight matrix based on a meta-codon for superimposed codons 3, 4, 5, 6, and 7 (see Figure 2).

### 4.2. Molecular Dynamic Analysis

Using the AMBER 18 suite [31], MD simulations were performed with a 494-residue subsystem of the ribosome centered around C1054 of the CAR surface, as described previously [18] except where noted below. A harmonic potential force (20 kcal/mol Å^2^) was applied to the residues at the surface of the subsystem (the onion shell) in order to retain the translocation stage II structure (PDB ID 5JUP [20]). The subsystem, based on cryo-EM structure 5JUP, includes parts of a viral IRES that mimic the tRNA and mRNA.

Editing of +1 codon nucleotides was accomplished by removing atoms not shared with the replacement nucleotide. Then, the AMBER tLEaP was used to “grow in” missing atoms for the new residue based on a standard geometry. Each tested +1 codon was characterized using 30 independent MD trajectories (twenty 60-ns trajectories and ten 100-ns trajectories). Each sample MD run was initiated with the same energy-minimized structure, which became an independent experimental run with random velocity assignment during heating, followed by equilibration before production MD. RMSD deviations of RNA and protein backbone atoms were monitored to ensure that the subsystem was stable, typically after 20 ns (Appendix A), and our subsequent analysis of the trajectories did not include these initial 20 ns.

H-bonding between CAR and the +1 codon as well as A-site codon/anticodon interactions were characterized using Python scripts to parse the results of the AMBER cpptraj [38] *hbond* function (*out* and *avgout* outputs), which returns H-bonds with an acceptor-to-donor heavy atom distance less than 3.0 Å and an angle cut-off of 135°.

A1196 stacking with C1054 (or R146) was characterized using the AMBER cpptraj *distance* function to measure the distances between the center of mass of the A1196 base with the C1054 base (or the R146 guanidinium group). Anchoring of C1054 to tRNA nt 34 through stacking was similarly measured. Moreover, we monitored water accessibility between the stacked groups by measuring the solvent accessible surface area (SASA) surrounding the two bases or the guanidinium group and base, using the fast LCPO algorithm [36] computed by the *surf* function in AMBER cpptraj. Furthermore, the geometries of base stacking interactions were visualized using a polar coordinate representation, as described by Bottaro et al. [32] to examine a single stacking pair of bases over multiple frames of trajectories—revealing centering of the stacked bases (ρ) and relative rotations of the base rings (θ) (Appendix A).

Root mean square fluctuation (RMSF) was measured using the AMBER cpptraj *rmsf* function on the base rings of RNA nucleotides and the guanidinium group of R146, as previously described [18].

### 4.3. Clustering Analysis of Trajectories

K-means clustering (Appendix A) was performed as described [37] using a concatenation of trajectories with different +1 codons. All of the frames from the concatenated trajectories were assigned to the cluster with their closest centroid structure based on RMSD measurement using the backbone atoms of all unrestrained residues in our subsystem (not including onion shell residues). Non-backbone atoms of nucleotides 1 and 2 of the +1 codon were removed allowing for a single parameter-topology (parmtop) file to be used for the trajectories with different +1 codons. K-means analysis was also performed using all atoms of 12 residues (A-site codon and anticodon, +1 codon and CAR). The resulting centroids provided representative structures for each cluster. These structures were viewed using Pymol [39] and virtual molecular dynamics (VMD [40,41]). To assess differences between structures in two clusters of interest, we performed RMSF on residues in one of the clusters using, as a reference structure, the centroid for the second comparison cluster. The RMSF values were visualized on the centroid structure of the first cluster by editing the B-factor column of its pdf file and visualizing in Pymol.

## Figures and Tables

**Figure 1 ijms-23-01417-f001:**
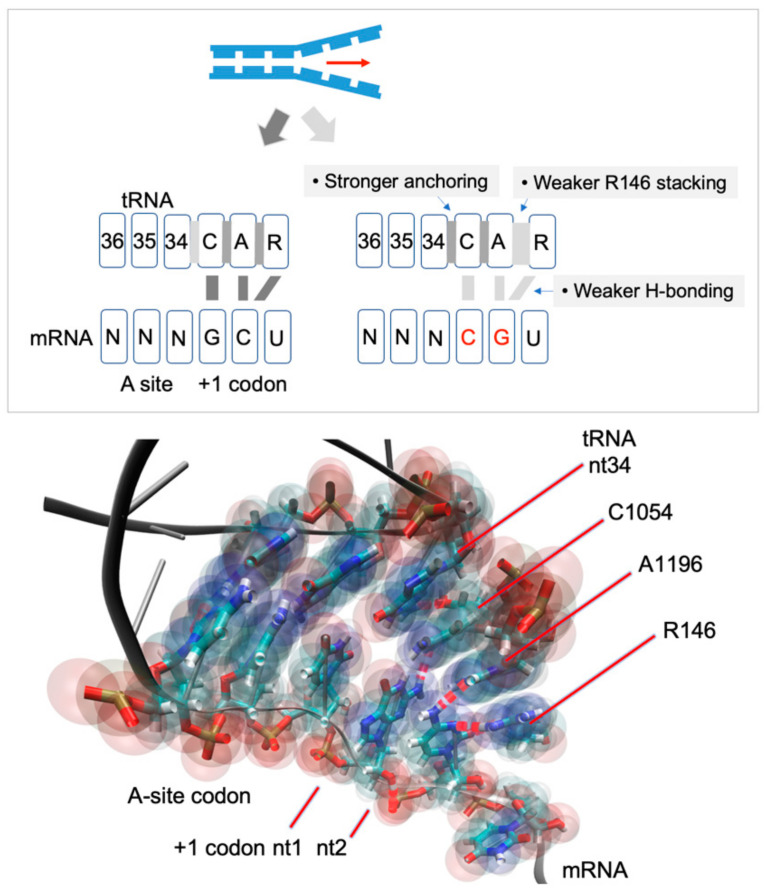
The ribosome CAR surface interacts with the mRNA +1 codon. The CAR surface is positioned close to the mRNA +1 codon allowing for H-bonding between CAR and the codon. Our study shows that this interaction is strongest with the +1 GCU codon. The weaker interaction with other +1 codons is associated with weaker stacking interactions between A1196 and R146, and stronger anchoring through stacking between C1054 and tRNA nt 34. The CAR/+1 codon interaction behaves like a zipper (red arrow; see text).

**Figure 2 ijms-23-01417-f002:**
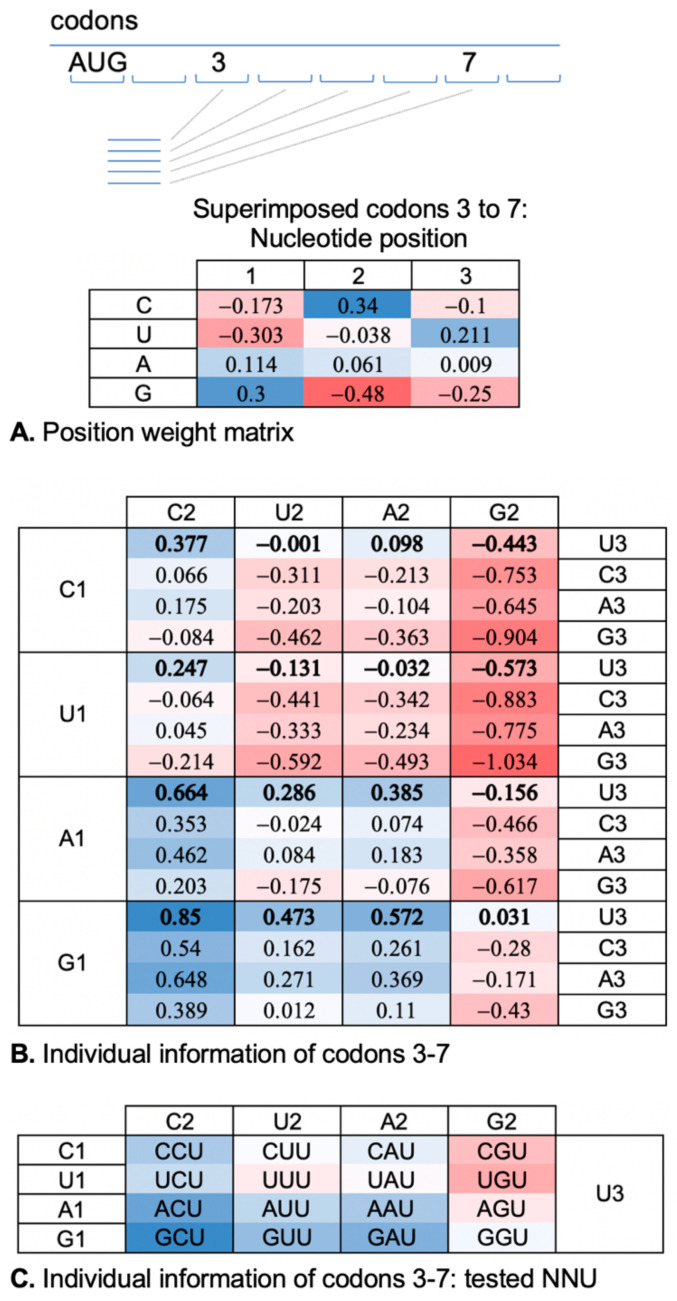
Information theoretic analysis of codons. (**A**) Codons 3–7 of ORFs of 6033 yeast genes were superimposed to create a meta-codon representation. The position weight matrix shows log_2_(f_observed_/f_expected_) for nucleotides at each codon position based on expected background frequencies in ORFs (A: 0.328, C: 0.192, G: 0.204, U: 0.276). (**B**) Individual information scores for codons were calculated using the position weight matrix. For example, the score for GCU is 0.30 + 0.34 + 0.21 = 0.85. (**C**) NNU codons were analyzed in this study with a heatmap color scale, indicating their relative individual information scores.

**Figure 3 ijms-23-01417-f003:**
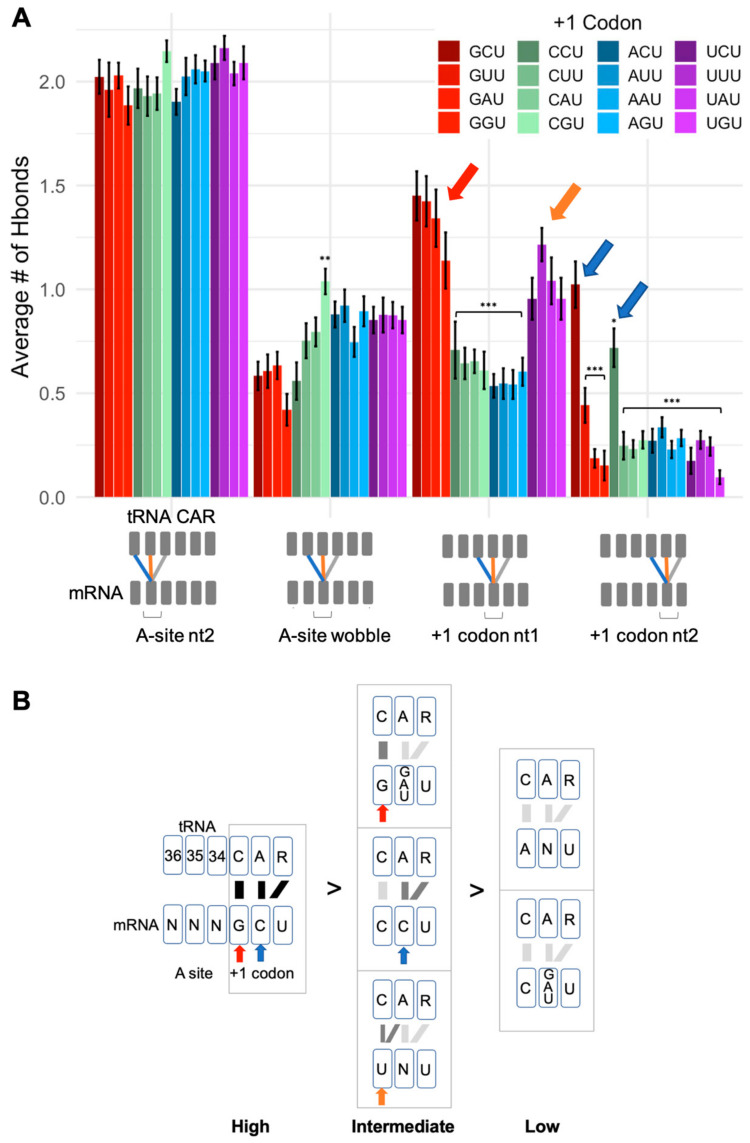
H-bond interactions of CAR. (**A**) Illustrated are H-bond counts between mRNA nucleotides (in the A site and +1 codon next to enter the A site) and the tRNA anticodon and ribosome CAR interface. Schematics show which pairs of residues are included in graphed H-bond averages. Statistical tests show comparisons with +1 GCU (*p* < 0.05 *; *p* < 0.01 **; *p* < 0.001 ***). Error bars show standard errors in H-bond counts from the MD. (**B**) Groupings of +1 codons with significantly different levels of H-bonding (*t*-test *p* < 0.001). In addition, +1 codons with G at position 1 (red arrows) or C at position 2 (blue arrows) have strong interactions. With U at position 1 (orange arrows), the H-bonding is split between C1054 and A1196 (see Figure 4).

**Figure 4 ijms-23-01417-f004:**
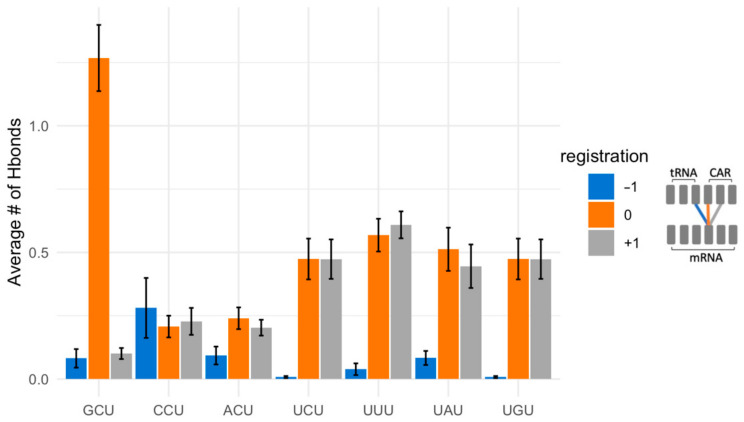
Alignment of CAR–mRNA interactions. Nucleotide 1 of the +1 GCU codon has dominant H-bond interactions with C1054. In contrast, the weaker interactions of U at position 1 (+1 UNN) are split between C1054 (orange) and A1196 (grey). C or A at position 1 has weak interactions with tRNA nt 34 (blue), C1054, and A1196.

**Figure 5 ijms-23-01417-f005:**
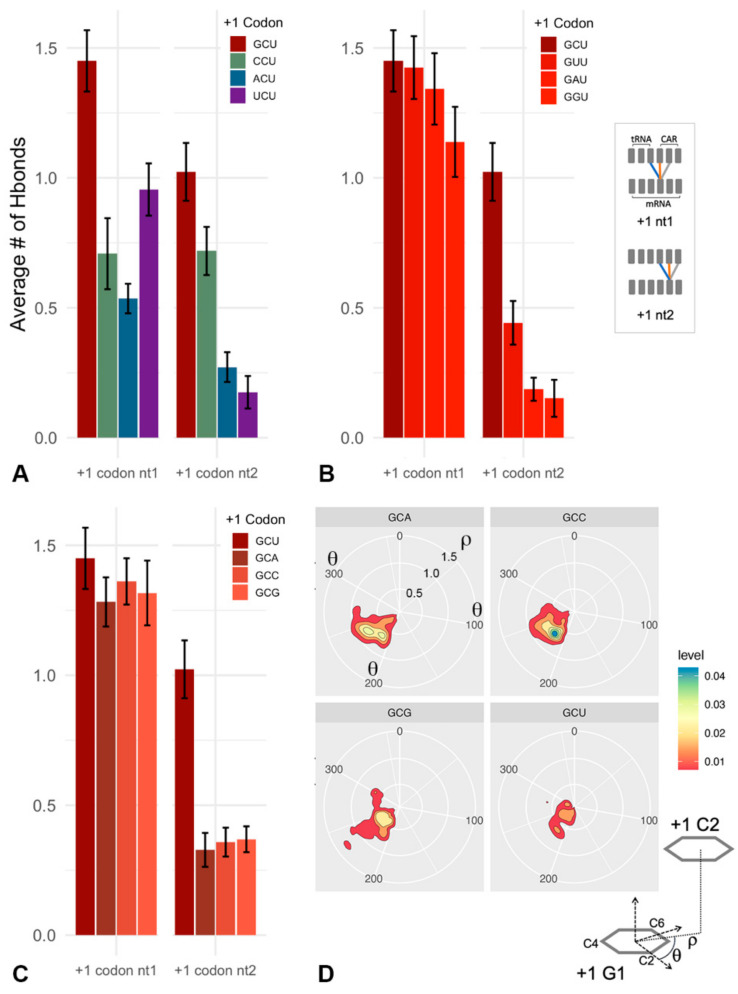
Influences of nucleotide positions on the CAR/+1 codon interaction. (**A**) When position 1 of the optimal +1 GCU codon is changed, both positions 1 and 2 of the codon have reduced H-bonding with CAR and tRNA nt34. (**B**) When position 2 of +1 GCU is changed, position 2 has significantly reduced H-bonding, but position 1 has minor reductions suggesting that the +1 codon interaction with CAR has zipper-like properties emanating from the A site. (**C**) When position 3 of +1 GCU is changed, the position 2 interaction with CAR is disrupted suggesting that position 3 may affect G1:C2 stacking. (**D**) A polar coordinate representation of G1:C2 stacking [32] across trajectories (1600 ns for each +1 codon) shows that G1:C2 stacking is less centered (smaller ρ-values) and more variable when U3 is replaced with A3, C3 or G3. Note that smaller ρ-values closer to the center have less area due to the polar coordinate projection onto a plane (see Appendix A).

**Figure 6 ijms-23-01417-f006:**
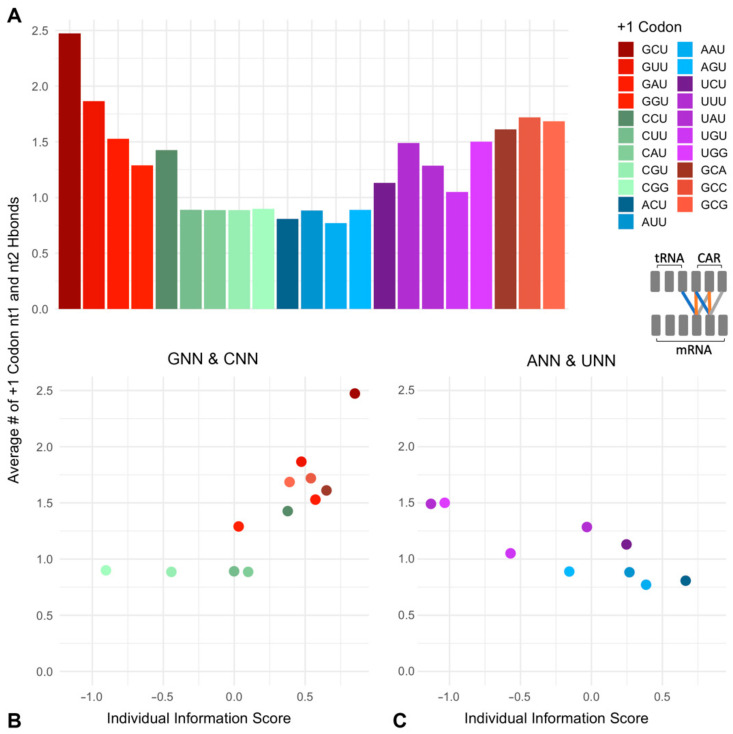
CAR interactions correlate with +1 codon individual information scores. (**A**) Summed H-bond counts for positions 1 and 2 of +1 codons. (**B**) Comparison of these H-bond counts with individual information scores of +1 GNN and +1 CNN codons revealed a correlation (Spearman correlation coefficient = 0.818). (**C**) However, +1 ANN and +1 UNN codons show a negative correlation (Spearman correlation coefficient = −0.833).

**Figure 7 ijms-23-01417-f007:**
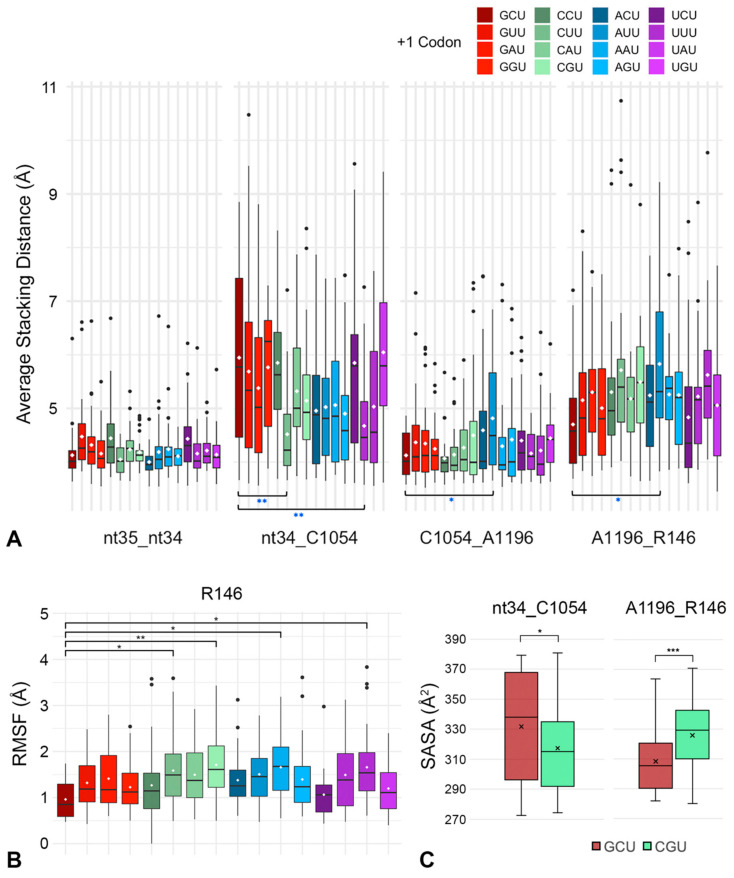
CAR stacking depends on +1 codons. (**A**) Stacking interactions in the CAR structure were assessed by measuring the center of mass (COM) distances between the base rings of C1054 and A1196, and between A1196 and the guanidinium group heavy atoms of R146. CAR anchoring was assessed through the COM distance between C1054 and tRNA nt34. The optimal +1 GCU codon had the strongest A1196_R146 stacking and weakest nt34_C1054 anchoring. (**B**) RMSF of R146 was correspondingly lowest for +1 GCU. (**C**) Solvent accessible surface area (SASA) measurements comparing +1 GCU and +1 CGU codons similarly showed that +1 GCU has lower SASA for A1196_R146 reflecting its stronger stacking, and higher SASA for nt34_C1054 reflecting its weaker stacking. White circles or “x” indicate means. Statistical tests show comparisons with +1 GCU (*p* < 0.05 *; *p* < 0.01 **; *p* < 0.001 ***).

**Figure 8 ijms-23-01417-f008:**
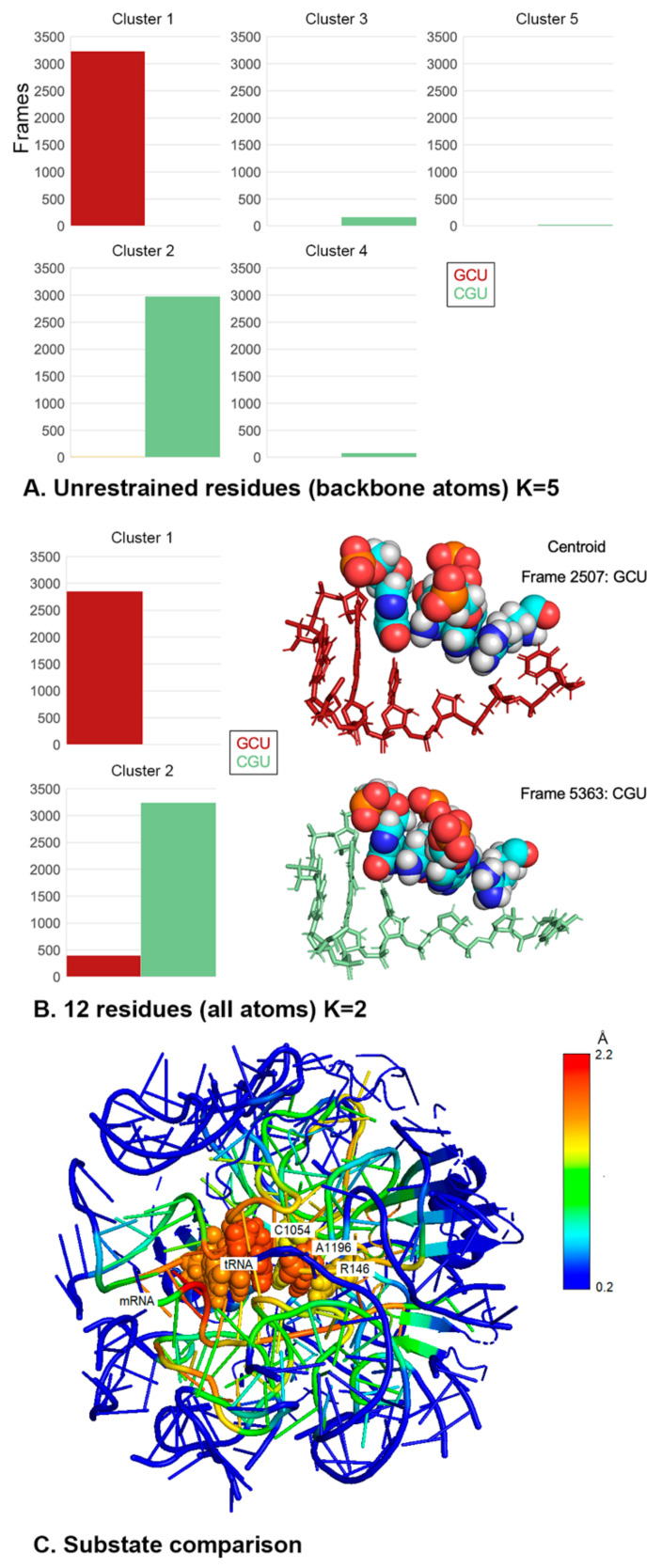
The decoding center adopts different backbone structures depending on +1 codons. (**A**) K-means clustering was performed on concatenated trajectories for +1 GCU and +1 CGU. Backbone atoms of the 321 unrestrained residues were used for clustering. Clustering with K = 5 revealed two dominant clusters, each populated by frames from only +1 GCU (cluster 1) or +1 CGU (cluster 2). In addition, 4.2% of +1 CGU frames were in other clusters. (**B**) K-means clustering was performed with all atoms of 12 residues (A-site codon and anticodon, +1 codon and CAR). Clustering with K = 2 revealed clusters dominated by +1 GCU or +1 CGU. The centroid frame structures are illustrated using van der Waals spheres for tRNA nt 34, C1054, A1196, and R146 (left to right). Cluster 1 (+1 GCU) shows weaker anchoring of CAR to tRNA nt 34, whereas the centroid for cluster 2 (+1 CGU) shows better CAR anchoring but poorer stacking of R146. The base rings of +1 codon nucleotides 1 and 2 are not illustrated and were not included in the clustering analysis. (**C**) Comparison of cluster 2 frames (CGU) with the centroid of cluster 1 (GCU, illustrated structure) from K-means analysis with backbone atoms (panel A) revealed extensive differences throughout the subsystem neighborhood (reds and yellows in structure heat map). As expected, the restrained onion shell residues show low RMSF values (blue).

**Figure 9 ijms-23-01417-f009:**
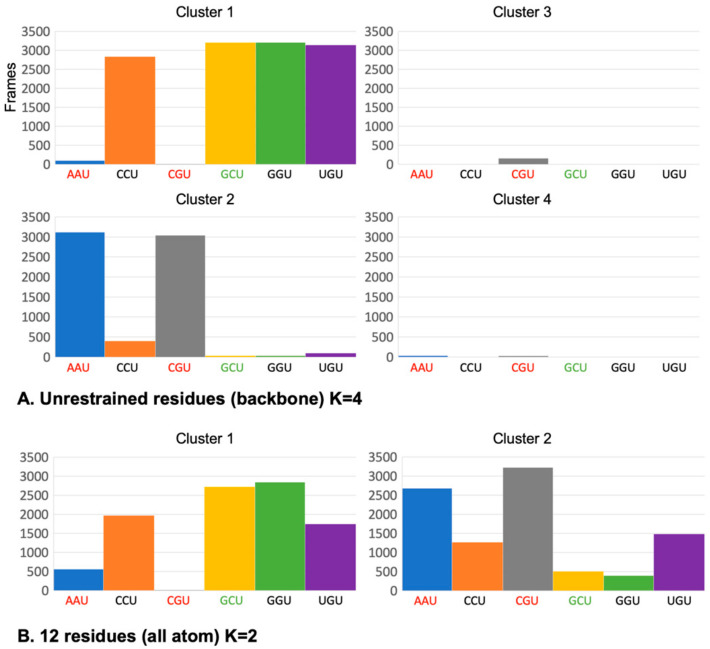
Trajectories for six +1 codons were concatenated for K-means analysis (30 trajectories each, two frames/ns). (**A**) Clustering (K = 4) with backbone atoms of unrestrained residues grouped together (cluster 1) frames for +1 codons with high (GCU) or intermediate (CCU, GGU, UGU) H-bonding between CAR and the +1 codon. Frames for +1 codons with low CAR/+1 codon H-bonding (AAU and CGU) were grouped in the other cluster (cluster 2), along with very small fractions of the CCU and UGU frames. (**B**) Clustering (K = 2) performed with all atoms of 12 residues (A-site codon and anticodon, +1 codon and CAR) revealed a cluster (cluster 1) dominated by GCU, GGU. In addition, the centroid frame for this cluster has weak anchoring of CAR to tRNA nt 34. The other cluster (cluster 2) is dominated by AAU and CGU, and its centroid frame (CGU, not shown) has good anchoring of CAR but weaker stacking of Rps3 R146.

## Data Availability

MD trajectory files are available from the authors upon request.

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
