# Peer review of "The CAR–mRNA Interaction Surface Is a Zipper Extension of the Ribosome A Site"

_ijms, 2022, doi:10.3390/ijms23031417_

Round 1

Reviewer 1 Report

With great pleasure, I have read and reviewed the article entitled: The CAR mRNA-Interaction Surface is a Zipper Extension of 2 the Ribosome A Site. In this article, the authors extend their previous hypothesis about ribosomes made under stress conditions and the meaning of CAR interface with unmethylated R146. Their careful structural analysis denoted that “in each of codon steps during ribosome translocation, the decoding center neighborhood adopts different substates depending on the identity of the +1 codon”. This observation put some light on translation speed and therefore cellular protein synthesis. Moreover, the knowledge of codon “story” is not sufficient in the nucleic acids damage context. From the editorial point, the article is well written and readable. The scientific sound is significant for a broad community. The materials and method have been presented properly in both cases bioinformatics and structural. I have two critical remarks: first, the part results and discussion is too long and should be shortened, second, figures 3 and 7 are difficult to read and must be proofed.

Author Response

The reviewer suggests that the Results and Discussion—which explores interactions between CAR and the +1 codon—could be shortened. We have removed several lines of text from Sections 2.1, 2.2, 2.4 and 2.5.

We agree that the previous versions of Figures 3 and 7 were difficult to read and have reorganized these figures to make the details more legible.

Reviewer 2 Report

The manuscript by Dalgarno et al. focuses on in silico study of the CAR mRNA interaction interface. Based on molecular dynamics simulations, the authors proposed translational regulation dependent on the CAR interaction surface, which is related to the identity of the +1 codon. The authors put effort into this work, and the calculations were performed in a detailed manner. The paper is well-written and, in my opinion, presented results are a good starting point for experimental (in vitro) study of such interactions.
Apart from the above, I suggest moving the sub-section 2.4. Conclusion to the new section 3.

Author Response

We agree with the reviewer that this in silico study of interactions of CAR with the +1 codon provides a good starting point for future experimental studies. As suggested by the reviewer, we have moved the Conclusions to Section 3.